# GWAS in a Collection of Bulgarian Old and Modern Bread Wheat Accessions Uncovers Novel Genomic Loci for Grain Protein Content and Thousand Kernel Weight

**DOI:** 10.3390/plants13081084

**Published:** 2024-04-12

**Authors:** Tania Kartseva, Vladimir Aleksandrov, Ahmad M. Alqudah, Mian Abdur Rehman Arif, Konstantina Kocheva, Dilyana Doneva, Katelina Prokopova, Andreas Börner, Svetlana Misheva

**Affiliations:** 1Institute of Plant Physiology and Genetics, Bulgarian Academy of Sciences, Acad. G. Bonchev Str., Block 21, 1113 Sofia, Bulgaria; tania_karceva@abv.bg (T.K.); aleksandrov@gbg.bg (V.A.);; 2Biological Science Program, Department of Biological and Environmental Sciences, College of Art and Science, Qatar University, Doha P.O. Box 2713, Qatar; aalqudah@qu.edu.qa; 3Nuclear Institute for Agriculture and Biology College, Pakistan Institute of Engineering and Applied Sciences (NIAB-C, PIEAS), Jhang Road, Faisalabad 38000, Pakistan; m.a.rehman.arif@gmail.com; 4Leibniz Institute of Plant Genetics and Crop Plant Research (IPK Gatersleben), OT Gatersleben, Corrensstraße 3, 06466 Seeland, Germany; boerner@ipk-gatersleben.de

**Keywords:** association mapping, candidate genes, grain protein content, TKW, grain quality, grain yield, *Triticum aestivum* L.

## Abstract

Genetic enhancement of grain production and quality is a priority in wheat breeding projects. In this study, we assessed two key agronomic traits—grain protein content (GPC) and thousand kernel weight (TKW)—across 179 Bulgarian contemporary and historic varieties and landraces across three growing seasons. Significant phenotypic variation existed for both traits among genotypes and seasons, and no discernible difference was evident between the old and modern accessions. To understand the genetic basis of the traits, we conducted a genome-wide association study with MLM using phenotypic data from the crop seasons, best linear unbiased estimators, and genotypic data from the 25K Infinium iSelect array. As a result, we detected 16 quantitative trait nucleotides (QTNs) associated with GPC and 15 associated with TKW, all of which passed the false discovery rate threshold. Seven loci favorably influenced GPC, resulting in an increase of 1.4% to 8.1%, while four loci had a positive impact on TKW with increases ranging from 1.9% to 8.4%. While some loci confirmed previously published associations, four QTNs linked to GPC on chromosomes 2A, 7A, and 7B, as well as two QTNs related to TKW on chromosomes 1B and 6A, may represent novel associations. Annotations for proteins involved in the senescence-associated nutrient remobilization and in the following buildup of resources required for seed germination have been found for selected putative candidate genes. These include genes coding for storage proteins, cysteine proteases, cellulose-synthase, alpha-amylase, transcriptional regulators, and F-box and RWP-RK family proteins. Our findings highlight promising genomic regions for targeted breeding programs aimed at improving grain yield and protein content.

## 1. Introduction

Wheat is a unique cereal crop due to the baking qualities of its flour and occupies a central strategic role in broadscale food security. Increasing yield and protein content in wheat grain is essential for safeguarding the human rights to sufficient and nutritious food. The quality and nutritional value of food products made from wheat flour largely depend upon the type and concentration of grain proteins [1,2]. As a consequence, total grain protein content (GPC) is one of the main determinants of both baking quality and the international market price of wheat [3]. The range of protein in wheat grains is usually between 8% and 20%, accounting for less than 8–15% of the grain dry weight. Thousand kernel weight (TKW) is a measure of grain weight and, along with spike number per unit area and grain number per spike, is a main component of wheat total yield. Grain weight and, respectively, grain yield is formed mainly by the starch accumulation in the developing grain and, therefore, any starch gain in the endosperm if not accompanied by an adequate rise in nitrogen (N)/protein accumulation has a dilution effect that influences the concentration of grain protein and micronutrients [4]. Due to this important interaction, breeders frequently encounter conundrums when aiming for genotypes that combine high yield and high protein content in the grain [5,6,7,8].

Comparative studies on changes that have occurred in wheat varieties released or introduced since the middle of the 19th century showed increased grain yield and decreased protein over time [4,9]. This suggests that old germplasm (landraces and traditional varieties) can be screened for new genetic diversity and targeted for the construction of new varieties. Old germplasm grown in Bulgaria has some desirable traits, such as consistent yield, resistance to drought, high protein content, or good quality for making bread [10]. The semi-dwarf high-yielding varieties that emerged in the 1970s replaced the landraces and the tall varieties that were previously grown [11]. Most of this germplasm is now extinct, but some seed samples are preserved and reproduced in the major European seed gene banks [12].

Both GPC and TKW are quantitatively inherited and are controlled by multiple genes or quantitative trait loci (QTL) [13,14]. The impact of the environmental conditions on gene expression and the genotype-by-environment interactions further complicates the precise evaluation of these traits [15].

Genome-wide association studies (GWAS) detect genetic effects based on linkage disequilibrium (LD) in natural germplasm collections and have become effective tools for modern plant breeding [16]. Following the rapid development of DNA marker technologies, and in particular the advent of single nucleotide polymorphism (SNP) chips, association mapping has been used progressively to establish a strong connection between a genome-wide SNP and a trait of interest. Identifying marker-trait associations (MTAs) or quantitative trait nucleotides (QTNs) can make it easier for breeders to choose the best genotypes, reduce the breeding cycle, and achieve higher genetic gains. A number of recent GWAS studies reported genomic regions associated with GPC and yield components [14,17,18,19,20,21,22,23,24,25,26,27,28,29,30,31].

The association panels employed in these studies varied in terms of diversity level, genetic relatedness, and the nature of accessions. For instance, Kartseva et al. [27] used a diverse population of 255 accessions from 27 countries on five continents, and revealed novel stable genomic regions harboring GPC-associated markers on chromosomes 3A and 3B. QTL hotspots containing 165 significant MTAs for quality and agronomic traits were mapped on almost all chromosomes in an association panel of 170 diverse landraces from the Mediterranean region [31]. Using a set of 93 spring common wheat varieties and breeding lines adapted for cultivation in the Siberian region of the Russian Federation, another study reported eleven genomic regions associated with GPC, of which nine were physically mapped on chromosome 6A harboring the *NAM-A1* gene, homoeologous to the *Gpc-B1* (*NAM-B1*) gene [23]. Another recent investigation by Tyrka et al. [30] screened 168 Polish breeding lines of common winter wheat for a number of agronomic traits, and identified trait-associated markers for heading time, lodging resistance, plant height, and TKW. Therefore, it is essential to evaluate new large mapping populations with different underlying substructure, extents of genetic relatedness among individuals, and LD decays to identify novel QTNs or QTL for yield and quality traits. Additionally, these traits are influenced by environmental factors; therefore, detecting MTAs in multienvironment or multiyear studies is crucial for their application in marker-assisted breeding.

For this study, we assembled an association panel of advanced and historical varieties and local accessions from Bulgaria. Our fundamental focus was to explore the natural genetic variation for protein content in wheat grain and thousand kernel weight, and to reveal the associated genetic determinants. We found a considerable amount of phenotypic variation for GPC and TKW with no apparent differences between modern high-productive varieties and old germplasm, nor did the traits significantly correlate with one another. Based on GWAS findings, we identified promising genomic regions for wheat improvement and uncovered possible candidate genes. These results provide information about the genetic resources available to breeders to improve grain yield and nutritional properties of wheat products, as well as an opportunity to develop closely associated markers to aid molecular breeding of new varieties.

## 2. Results

### 2.1. Phenotypic Variation

Descriptive statistics, frequency distribution, and boxplots showed wide phenotypic variation for both GPC and TKW over three growing seasons (harvests 2014, 2017, 2021) within the set of 179 wheat accessions with total average over the crop seasons of 13.4 ± 1.34% for GPC, and 45.2 ± 4.28 g for TKW (Table 1, Appendix A; Figure 1). The range of coefficients of variation (CV, %) across the individual years was similar for the two traits. Estimates of high broad-sense heritability (*h*^2^) for both traits showed moderate to high values in the individual growing seasons (from 0.64 to 0.78), and high values over the environments (0.82 for GPC and 0.81 for TKW). To exclude the impact of the growing season, we computed best linear unbiased estimator (BLUE) values for each accession, treating genotype as fixed and growing season as random effects. BLUEs varied across the years from 11.6 to 14.7%, on average 13.4% for GPC, and from 39.5 to 51.8 g, on average 45.4 g for TKW (Table 1, Appendix A; Figure 1).

The statistical analysis (Shapiro–Wilk test) indicates that the phenotypic data adheres to a normal distribution at a *p*-value threshold of 0.05. The ANOVA results explained the presence of broad phenotypic variation among genotypes for GPC and TKW, revealing highly significant effects of genotype, environment (growing season), and their interactions (Table 2). For both traits, no significant differences were noted only between 2014 and 2017 (Figure 1, Appendix A).

To assess the trait consistency across the environments and to explain the relationships between GPC and TKW, Pearson’s correlation coefficient approach was used. Low to high positive Pearson’s correlation coefficients (*r*) over the years were computed for GPC (spanning from 0.50 to 0.92) and TKW (ranging from 0.30 to 0.79) (Table 3). In general, consistency was noted across the growing seasons, with one exception—GPC in 2017 was not correlated with that in 2021. From the perspective of the data desirability for GWAS, correlation analysis was performed also with the BLUE mean values. The Pearson’s correlation coefficients (*r*) computed based on BLUEs were positively significant (*p* < 0.001) for both GPC and TKW. The two grain characteristics were not correlated across the growing seasons, as well as based on BLUEs (Table 3).

According to a previous work, the studied population has a distinct structure based on the membership coefficient matrix (Q-matrix), calculated in STRUCTURE [32]. It is composed of three subpopulations (SPs), and only five genotypes are deemed to be admixed. The largest subpopulation—SP1 (109 entries) includes 6 old accessions and 103 modern varieties, most of them developed in the breeding centers in the Northern and Southern parts of the country. Of the 49 accessions in the SP2 subpopulation, 43 belong almost solely to the old germplasm, with the majority of them originating from Northern Bulgaria. The remaining 6 accessions are contemporary varieties. The smallest subpopulation—SP3—consists mostly of modern varieties (14), and 2 old accessions. Taking into account the distinct population structure, we tested the hypothesis of whether the old accessions differ significantly and consistently from the modern releases. Therefore, we compared the three SPs across the growing seasons. While the two subgroups containing modern varieties did not differ significantly across the years and with BLUEs concerning both GPC and TKW, the subgroup of old accessions (SP2) displayed variability across the environments. Subpopulation SP2 had significantly lower GPC compared to the SP3-varieties in 2021, but did not differ from both groups of modern releases in 2014 and 2017 and with BLUEs (Figure 2a). The group of old germplasm showed a higher TKW mean value in 2017 but a lower value in 2021 when compared to SP1, whereas the TKW mean BLUE of SP2 was significantly lower than that of SP3 (Figure 2b).

To distinguish genotypes that could be used for improving grain protein content in breeding programs, the accessions were classified according to [2] into five groups with protein ≥ 13% (Group 1), ≥12% (Group 2), ≥11% (Group 3), and >10% (Group 4), <10 (Group 5). The prevailing part of the accessions (90% and 94% of SP1 and SP3 modern releases, 84% of SP2 old accessions, or 89% of the entire population) fall into Groups 1 and 2 (Figure 3).

In order to evaluate the phenotypic variability of the population throughout the crop seasons, we initially determined the deviations from the respective yearly average GPC and TKW. Subsequently, these discrepancies were averaged for each accession throughout the three seasons. The influence of genotypes fluctuated per the environmental gradient, with the variance extending from 0.012 (2014) to 0.025 (2017) for GPC (Appendix A), and from 0.011 (2017) to 0.022 (2014) for TKW (Appendix A). The variance within the entire population over the years spanned from 0.0 to 0.108 for GPC, and from 0.0 to 0.101 for TKW.

### 2.2. Linkage Disequilibrium (LD) Estimation, Genotype Relatedness, and Significant Quantitative Trait Nucleotides (QTNs)

From the perspective of GWAS, LD was considered. This is essential to define the interval of highly associated SNPs and to identify the most significant loci [16]. Here, we calculated LD decay using the whole association panel irrespective of the genotype status (old vs. modern). The LD decay values varied from 1.5 to 3.0 Mbp on the individual chromosomes, with the highest value of LD decay in the D-genome (2.54 Mbp) and homoeologous group 3 (2.27 Mbp), on average 1.98 Mbp (Appendix A).

High relatedness between the genotypes was demonstrated by the phylogenetic tree and the heat map of the values in the kinship matrix constructed from 17,083 SNPs (Appendix A).

In GWAS, controlling for population structure is a routine method that is especially relevant in this study because the genotypes comprise both historical and contemporary varieties as well as landraces. In an earlier study, the degree of diversity and population stratification were examined [32]. Three subpopulations were identified by STRUCTURE and the k-means clustering algorithm. The old germplasm was distinguished by both methods, as well as by principal component analysis. This work provides also a comprehensive comparative analysis of the old and modern accessions in terms of SNP-based genetic diversity and LD [32].

For both traits, the phenotypic data from three crop seasons and the genotypic data for the association panel that were already available from the 25K Infinium iSelect array (SGS Institut Fresenius GmbH TraitGenetics Section, Gatersleben, Germany) and described in [32] were used for association mapping. An examination of the variance showed substantial genotype-by-environment interactions (Table 2), despite the fact that both traits showed moderate to high broad-sense heritability estimates in each crop season (Table 1). Therefore, association mapping analysis was performed for each crop season, followed by analysis based on BLUEs using MLM (Q + K) model to account for population structure and kinship. The Q-Q plots showed deviations of the observed *p*-values from the null hypothesis suggesting there is a certain overestimation of the positive genomic signals (Appendix A). Therefore, in order to maintain a low false positive rate, the significance threshold was set at −log_10_ (*p*-value) > FDR (false discovery rate), *p* < 0.01, and was calculated for each trait and environment.

GWAS for GPC evidenced 16 significant QTNs (−log_10_ > FDR, *p* < 0.01) distributed on 11 chromosomes across the three environments and BLUEs: 1B, 2A, 2B, 4B, 5A, 5B, 6A, 6B, 6D, 7A, and 7B (Table 4, Figure 4a). For the yield-related trait TKW, a total of 15 significant QTNs distributed on chromosomes 1A, 1B, 2A, 4A, 4D, 5B, 6A, and 6D were detected over the three crop seasons and based on BLUEs (Table 5, Figure 4b). The proportion of phenotypic variance (R^2^) explained by the significantly associated markers was within the range 0.09 to 0.27 for GPC, and 0.12 to 0.23 for TKW; this range may have been upward biased to the relatively small population size. Seven QTNs presented positive additive effect on protein content spanning from 1.4% (*RFL_Contig5739_641*) to 8.1% (*AX-158545828*). Regarding TKW, four markers were identified as having positive additive effects on the trait, with *wsnp_JD_c4217_5322858* having the most impact (8.4%).

No environmentally stable QTNs were detected for GPC and TKW, which confirms that both grain traits are significantly influenced by the crop seasons (environments) and demonstrate the presence of genotype-by-environment interactions.

In our analysis, we examined the significant markers associated with GPC and TKW across the three distinct subpopulations, as differentiated by the Q-matrix in a previous study [32]. Our observations indicate that among the 16 loci significantly associated with GPC, only locus *BS00109912_51* exclusively (100%) carries allele C in the old accessions (SP2 subpopulation). However, the other allele (T) is still present in the old germplasm, albeit at a lower frequency (20% of all accessions with this allele) (Appendix A). In the case of 8 out of 15 significant loci associated with TKW, one allele is predominantly (95–100%) present in the SP1 and SP3 subpopulations, which are mainly composed of modern varieties. The other allele is also highly represented among the modern releases (Appendix A).

### 2.3. Potential Candidate Genes

In order to hypothesize potential candidate genes that underlie the analyzed traits, we examined the 16 significant QTNs detected for GPC and the 15 significant QTNs detected for TKW. The search for candidate genes was performed based on the physical position of the prominent QTNs on *Triticum aestivum* L. cv. Chinese Spring reference genome extended by the LD interval estimated for each chromosome based on LD decay. Within the defined regions, associated with GPC and TKW, we retrieved a total of 493 and 639 high-confidence annotated genes, respectively (Appendix A). These genes were evaluated as potential candidate genes. The genes directly hit by the significant SNPs, along with their putative functions, are presented in Table 4 and Table 5.

In the following section, we look at the genes with significant QTNs of positive additive effects, as well as at several pertinent candidates detected in the LD-defined intervals on both sides of the QTNs. Relevant potential genes linked to GPC and TKW encode storage proteins, proteins implicated in senescence- and germination-associated proteolysis, carbohydrate synthesis, protein synthesis and trafficking, and auxin biosynthesis. Products of certain candidate genes are transcriptional and posttranscriptional regulators, or transmembrane factors that can sense, transduce, and transmit signals.

## 3. Discussion

### 3.1. Phenotypic Variation

To meet the growing demand for sufficient amounts and quality of food and overcome the challenges posed by environmental changes, it is essential to effectively utilize the available genetic resources of bread wheat [39]. Here, we used a collection of old and modern bread wheat accessions, and explored the genetic variation of TKW and GPC, the two most important characters determining yield and end-use quality, and eventually, the economic value of bread wheat. The study revealed large phenotypic variability for the target traits with highly significant contributions of genotype, environment, and genotype-by-environment interactions. The observed moderate to high values of broad-sense heritability and the consistency of trait records evidenced by the significant correlations between the crop seasons suggest that a considerable part of the variation is due to inherent genetic differences among the accessions. These results agree with similar findings in bread and durum wheat for GPC [19,25,27,40], and TKW [13,41]. Given that the panel contains historical and contemporary varieties released or collected between 1925 and 2010, it is possible that the recorded genetic variance—especially for TKW—is inflated. This is because throughout the course of nearly nine decades, genetic factors have contributed significantly to increases in yield and yield-related attributes. The landmark of these genetic gains for the Bulgarian wheat collection is the introduction of semi-dwarfing genes in the 1970s [11]. The pleiotropic effects that these genes exert on plant responsiveness to N applications, on photosynthetic rates, and on the accumulation of carbohydrates in the grain ultimately have improved grain yield.

This study shows that a large proportion (89%) of the accessions have protein above 12%, and are suitable for preparing leavened breads [2]. The average grain protein values (13.4%) are similar to or higher than the reported information on protein contents in other bread wheat collections [19,27,28,31,34]. Additional research revealed that some accession sets had greater protein values (up to 20%), demonstrating the impact of genotype-by-environment interactions [22,23,33].

Wheat yield and quality are affected by climatic factors (temperature, precipitation, drought type) [42]. In contrast to the abundant rainfall that was recorded at the experimental site from April to July in 2014, stretching from heading time to maturity, the amount of rain that fell in May and June 2017 was consistently less than the average for the climate (Appendix A), which suggests a sustained moderate drought during anthesis and grain filling. Prolonged water insufficiency early in grain development reduces the number of amyloplasts and endosperm cells, which lowers the capacity of starch accumulation, and ultimately lowers grain weight [43,44]. In 2021, a modest drought during anthesis and a more severe final drought were seen (Appendix A). Minor postanthesis water stress can speed up grain filling in wheat by boosting the activity of key catalytic enzymes that convert sucrose to starch, and by remobilizing nonstructural carbohydrates from the vegetative tissues to the grain [44]. The effects of these patterns of drought (timing and severity) explain well the significantly higher mean TKW value obtained in 2021 than those in 2017. The considerably lower mean TKW recorded in 2014 is consistent with research showing that prolonged soil wetness after anthesis restricts the amount of assimilates available to growing grains, hence diminishing the development of grain yield [45]. The observed variance in the protein content in the collection under study may potentially be explained by the precipitation oscillations among the three crop seasons. In our study, we found a significant difference (*p* < 0.001, Table 1) in the average protein values for 2014 (13.2%) and 2021 (14.2%), when prolonged waterlogging or terminal drought periods, respectively, were suggested during late vegetation. A decline in GPC has been associated with waterlogging [46]. Conversely, during a drought, an increase in the seed proteins has been shown [47], possibly connected to modified carbon (C) partitioning and, hence, to a shift in the C/N ratio, that favors greater N-assimilation [48].

Significant genotype-by-environment interactions were seen (Table 2), and the genotypic effects varied along with the environmental gradient (Appendix A). It is interesting to note, that throughout the growing seasons varieties Mustang, Bozhana, and Levent displayed consistently high positive deviation from the average GPC and high values of TKW. These accessions are potential sources for concurrent improvement of the two traits. Ancestral history indicates that the Mustang variety is descended from the old accession Yubilejna-3, which also exhibits consistently high protein levels in the grain. Additionally, its pedigree involves a hybrid derived from *Agropyron* sp., a wheatgrass that has been shown to contribute to seed storage proteins in wheat-*Agropyron* introgressions [49,50].

The lack of association between TKW and GPC in our data using Pearson’s approach shows that there is little to no decrease in seed weight in the presence of the identified loci for GPC. Similarly, no significant correlations were found between GPC and TKW across environments in studies of wheat lines derived from wild emmer [51,52]. This observation is consistent with findings by Oury and Godin [6] that genotype-by-environment interactions for grain yield and GPC may obscure the strong genetic background of the yield–protein interrelationship. Thus, Lindeque et al. [53] detected limited significant correlations between grain yield and protein content in a study of wheat accessions of various yielding capacities grown in environments of different precipitation trends. This result holds promise for the simultaneous genetic enhancement of the two traits.

Historical germplasm, such as landraces and traditional varieties, are an important source for bringing new genes into contemporary crops [31,54,55]. Therefore, it is strategically important to characterize these genetic resources in order to properly utilize them in prebreeding. Our comparative analysis of variation for GPC showed that although the old germplasm (SP2) was characterized in general by high values of grain protein concentrations (Figure 3, Appendix A), these values were lower than those of modern releases or did not differ notably from them (Figure 2). Moreover, the percentage of high-protein genotypes (≥12%) was higher in the two modern subpopulations, SP1 (90%) and SP3 (94%), compared to the old accessions of SP2 (84%). The idea that the older germplasm had superior yield-related metrics than the more recent varieties was also not supported by the observed variability in mean TKW values of SP2-accessions across the growing seasons. One likely reason is that a lot of contemporary Bulgarian varieties are either descended from old accessions, or have highly productive and high-protein Russian and Serbian ancestors in their pedigrees as Appendix A and the research by Kartseva et al. [27] show. These results suggest that wheat breeders have managed to construct improved varieties in terms of the studied traits.

### 3.2. Genomic Regions Associated with Grain Protein Content (GPC) and Thousand Kernel Weight (TKW)

Significant verified SNPs or genomic regions linked to GPC and TKW have been detected on all wheat chromosomes, according to recent research [14,35,56,57]. We compared the strongly associated SNPs found in this study to previously published loci or markers using the IWGSC RefSeq v.1.0 map as a reference.

Our analysis confirmed some known trait-associated genomic regions, or found associated markers in proximity (distance 0.1–6.5 Mbp) to previously described loci [21,23,33,34,35,36,37,38] (Table 4 and Table 5). For example, the GPC-associated marker *Tdurum_contig46670_911* on chromosome 6A (position 599,050,921 bp) has been reported as an environmentally specific one in a study on a Siberian wheat collection [23]. The same marker is also located inside a known QTL (q6A-3), defined by the interval 599.2–602.9 Mbp [33], and stays two Mbp apart from locus *QGpc.ipg-6A.4* reported by Leonova et al. [23] (Table 4). Similarly, six TKW-associated QTNs detected on chromosomes 1B, 5B, 6A, and 6D overlapped with previously reported QTL [21,35,37] (Table 5).

Certain significant QTNs have not been previously linked to the studied traits, suggesting that these loci are novel. New genomic loci that seem promising for grain protein improvement in wheat harbor markers *Ra_c22880_760* and *AX-158545828* on chromosome 2A, *AX-94519723* on chromosome 7B, and *BS00109912_51* on chromosome 7A, presenting protein increasing effects up to 8.1%. Two potentially novel loci with positive additive effects on TKW are defined in this study. The linked markers are *Tdurum_contig50667_306* on chromosome 1B, and *AX-95145282* on chromosome 6A. Some of the trait-associated loci were detected with BLUEs, whereas the rest are environmentally specific and need further validation.

The comparative analysis of the historical and contemporary genotypes, focusing on the significant markers associated with GPC and TKW, revealed little to no divergence of the old germplasm. This observation is consistent with molecular findings from previous research on Bulgarian wheat germplasm which utilized microsatellites [58] and SNPs [32]. These studies demonstrated that wheat breeders have maintained the high levels of genetic diversity found in the older germplasm since the 1970s. Furthermore, a shift in some prevailing alleles from the older accessions to the modern ones seems to have occurred, with some unique or rare alleles disappearing and others being introduced.

### 3.3. Putative Candidate Genes Related to Grain Protein Content (GPC) and Thousand Kernel Weight (TKW)

Grain production and nutrient content of cereal crops are governed by the buildup of nutrients in the grain near the end of the plant lifespan [59]. The quantity and quality of these nutrients is greatly affected by the remobilization of C and N from vegetative tissues to developing grain during senescence, as well as by the subsequent accumulation of resources for seed germination. All these processes are accompanied by expressional changes in a vast number of genes that ultimately impact grain output and protein content [60]. We searched for potential candidate genes at suggestively novel loci, as well as at loci that verified previously established associations. The next section covers potential genes in genomic regions harboring QTNs with positive additive effects on the traits.

Our findings highlighted seven QTNs with positive effects on grain protein, of which the two QTNs with the highest effect (ca. 8% each) are located in the interval 498,105,752–501,850,514 bp on chromosome 2A. Marker *Ra_c22880_760* resides within *TraesCS2A01G289800*, a gene translated into alpha-amylase. Alpha-amylase is one of the primary enzymes responsible for starch degradation to fuel early germinating grain [61]. The gene *TraesCS2A01G291200* hit by marker *AX-158545828* is annotated as mitochondrial import inner membrane translocase subunit TIM22. This protein is a PRAT (preprotein and amino acid transporters) family member and an important constituent of the mitochondrial protein import machinery [62]. Although the effect of this gene has not previously been documented in wheat, the functions of the *Arabidopsis* orthologous gene during seed development have been recently reported [63].

The gene *TraesCS7B01G105700* on chromosome 7B hit by marker *AX-94519723* encodes for a DNA-binding storekeeper protein-related transcriptional regulator (STKR). The STKR was the first B-box motif plant-specific binding protein identified as a putative regulator of expression of patatin, the most abundant storage protein of potato tubers [64]. In transgenic *Arabidopsis* plants, overexpression of the orthologous protein (STKR1) was shown to result in reduced growth, a delay in flowering, attenuated senescence, exhausted carbohydrate pool, and amino acids accumulation [65] and, therefore, is related to nutrient source-to-sink remobilization.

One QTN (*BS00109912_51*) is located on chromosome 7A, possibly in a noncoding interval. Within the LD-expanded genomic region to the left and right of the significant SNP, we retrieved two genes encoding cysteine-proteases. These enzymes are among the most abundant proteases activated during leaf senescence that are essential for N remobilization from senesced leaves to developing seeds [66].

We found prospective putative candidate genes for GPC improvement in the genomic region on chromosome 6A harboring marker *Tdurum_contig46670_911* (position 599,050,921 bp). The gene *TraesCS6A01G378100* translated into Diphthine-ammonia ligase has not been previously associated with effects in wheat. This enzyme is evolutionarily conserved in eukaryotes and catalyzes the last step in the conversion of an L-histidine residue in the translation elongation factor eEF2 to diphthamide [67]. Interestingly, there is a gene close to *TraesCS6A01G378100*, encoding translation elongation factor EF1A. The elongation factors (EFs) are key mediators in the peptide chains elongation during protein synthesis [68]. Nearby *TraesCS6A01G378100* we found two genes encoding NAC-domain-containing proteins. *NAC* genes, especially *NAM-B1* (*Gpc-B1*) together with its homoeologs *NAM-A1* and *NAM-D1* have been shown to regulate the transcriptional changes during leaf senescence in wheat, affecting N and nutrient source-to-sink remobilization [69,70]. Marker *Tdurum_contig46670_911* identified in the current study is situated 0.1 Mb away from a GPC locus (*QGpc.icg-6A.4*), described in the distal part of the long arm of chromosome 6A [23]. Since *NAM-A1* gene is located on the short arm of 6A [23], the *NAC* genes identified here close to *TraesCS6A01G378100* are, therefore, distinct.

The QTN *RFL_Contig5739_641* showing increase on GPC of ca. 1.4% resides inside the gene *TraesCS5B01G350900* coding for plant regulator RWP-RK family protein. RWP-RKs represent a small family of transcription factors that are unique to plants. The NLPs (NIN-like proteins) subfamily of RWP-RKs have been confirmed as master regulators of the nitrate signaling and the expression of a series of genes for nitrate transport and assimilation, including nitrate-transporter and nitrate reductase genes [71,72]. Interesting putative candidates for grain protein are four genes detected nearby *TraesCS5B01G350900* that were translated into flavin-containing monooxygenase, and one gene coding for subtilisin-like protease. In plants, the flavin-containing monooxygenase (FMO) gene superfamily is implicated in the auxin biosynthesis [73]. Auxin plays a pivotal role as a regulator, modulating a variety of cellular processes that are integral to seed formation. Genes coding for auxin-related FMO had steadily increasing expression levels during the wheat grain expanding phase, indicating a potential role in grain development [74]. Subtilisin-like proteases contribute to the degradation of reserve proteins during germination [75]. Genes for subtilisin-like proteases have been proposed as possible candidates for GPC [19] and seed longevity [76] in wheat.

Another QTN, which contributes to approximately a 1.4% increase in protein, is located within the gene *TraesCS6B01G052900* on chromosome 6B. This gene codes for a Kelch repeat–containing protein, a subclass of F-box proteins. F-box proteins are important players in the ubiquitin/26S proteasome system [77]. Protein breakdown, one of the most significant catabolic processes associated with leaf senescence, plays a key role in nutrient recycling, especially N. Consequently, it is closely linked to grain filling in cereals [75]. Chen et al. [78] reported on an F-box protein containing a Kelch repeat motif (OsFBK12) that regulates leaf senescence, seed size, and grain number in rice (*Oryza sativa*). The expression of wheat *TaKFB* genes encoding Kelch repeat F-box proteins was shown to increase during the pigmentation stage of grain development [79]. Close to *TraesCS6B01G052900* we found another gene coding for subtilisin-like protease, discussed above.

Four QTNs were identified that had positive effects on TKW. The QTN on chromosome 4A with the highest effect (ca. 8%) co-locates with *TraesCS4A01G115300*, a gene encoding small nuclear ribonucleoprotein (snRNP). The snRNPs, along with over 300 other splicing factors, form the spliceosome, the enzyme that catalyzes the splicing of the primary transcripts to mature mRNAs, an essential step in gene expression [80]. Lopato et al. [81] showed that the wheat gene *TaRSZ38*, a homologue of the *Arabidopsis At*RSZ33 splicing factor, regulates posttranscriptional processes in wheat grain with an expression peak in the immature grain. A relevant candidate gene for TKW, located 682,883 bp apart from *TraesCS4A01G115300* is translated into aminotransferase. A genome-wide analysis in bread wheat has identified 15 Tryptophan Aminotransferase of Arabidopsis1/Tryptophan Aminotransferase-Related (*TAA1/TAR*) genes involved in auxin biosynthesis. These findings highlight the potential of the gene *TaTAR2.1-3A* for genetically enhancing grain yield. It has been demonstrated that overexpression of this gene leads to increased plant height, spike number, grain yield, and aerial N accumulation under various N supplies [82].

Our analysis did not detect a candidate gene directly associated with QTN *AX-95145282* (position: 100,766,008 bp, chromosome 6A) having ca. 5% increasing effect on TKW. The gene search within the LD-expanded interval found 16 genes encoding leucine-rich repeat receptor-like protein kinase family protein (LRR-RLKs) and a squamosa promoter-binding protein-like (SBP domain) transcription factor family protein. LRR-RLKs are the largest group of receptor-like kinases in plants implicated in plant growth, development, and stress responses [83]. The effects of this protein family on yield-related traits in wheat have not been reported, but overexpression of the rice gene *LRK1* has been shown to increase cell proliferation and several yield components, resulting in a 27% increase in total grain yield per plant [84]. Studies in rice [85] and wheat [86] reported on putative candidate genes for grain size and TKW, encoding for squamosa promoter binding protein-like (SPL), a star player for plant growth and development [87]. Interestingly, the *Arabidopsis AtSPL9* gene was reported as a potential regulator of nitrate transporters and nitrate reductase [72] thereby possibly regulating the N assimilation.

The QTN *Tdurum_contig50667_306*, located on chromosome 1B (position: 20,588,032 bp; ca. 3% trait increase) is inside the gene *TraesCS1B01G041200*, that translates into an F-box protein-like. The role of this super protein family in plant cells was briefly covered previously in this section. Recent studies suggested that the wheat gene *WAPO-A1*, also known as *TaAPO-A1*, encoding an F-box protein is associated with variation for grain yield and quality traits [8,88,89]. In close proximity to *TraesCS1B01G041200*, we found another gene that seems to play a significant role in explaining the variation for TKW. This gene encodes a developmentally regulated G protein. Heterotrimeric G proteins in plants are membrane-attached signal transducers known to control diverse cellular processes, such as cell proliferation, hormonal responses, growth, and development [90]. G protein-mediated regulation of grain size has been reported in *Arabidopsis* [91], rice [90,92], barley [93], and could therefore impact the harvestable yield in crop plants. Moreover, the plant G protein complex was shown to regulate N signaling and N use efficiency in rice [94].

The QTN *AX-94692394* on chromosome 1A with the lowest increasing effect on TKW (ca. 2%) resides within gene *TraesCS1A01G012600,* translated into leucine-rich repeat receptor-like protein kinase family protein. Earlier in this section, we provided short information about the possible functions of LRR-RLKs. Interestingly, we discovered a putative candidate gene, *TraesCS1A01G010900*, in the LD interval close to *TraesCS1A01G012600* annotated as gliadin/LMW glutenin/bifunctional inhibitor/plant lipid transfer protein/seed storage helical domain. Similarly, Giancaspro et al. [86] and Schierenbeck et al. [95] identified candidate genes coding for the primary prolamins in TKW-coding regions. This finding is not surprising given that gliadins and glutenins make up the majority of endosperm reserve proteins in wheat grain. Others also reported on such function of suggestive candidate genes underlying marker associations with grain architecture traits and TKW [36,95]. Another relevant candidate in the 1A-genomic region (*TraesCS1A01G010500*) encodes cellulose synthase-like protein (Csl). This enzyme controls the cellulose biosynthesis thereby contributing to the cell wall organization. A recent genome-wide analysis of the *Csl* gene family in bread wheat revealed 108 *Csl* genes, and some of them were expressed in the grain [96]. Interestingly, a candidate gene for the wheat tiller inhibition gene (*tin*) that reduces tillering and increases grain weight was predicted to encode a Csl protein [97].

## 4. Materials and Methods

### 4.1. Plant Material

The plant material consisted of 179 winter wheat (*Triticum aestivum* L.) accessions that originated from Bulgaria, of which 129 were modern varieties, and 50 were old accessions (historic varieties of tall stature and landraces). Seeds from the old accessions were procured from the seed gene banks at the Leibniz Institute for Plant Genetics and Crop Research (IPK), Gatersleben, Germany, and the Crop Research Institute, Prague, Czech Republic. According to the available information, the period of early breeding and research expeditions to aggregate seeds from landraces and historic varieties for gene bank collections expands from 1925 to 1970. Seeds from the contemporary varieties released until 2010 were made available from the two major breeding centers in Bulgaria (Dobrudzha Agricultural Institute, General Toshevo and the Institute of Plant Genetic Resources, Sadovo, Bulgaria), as well as by breeders. Information about botanical variety, status (modern vs. old), year of release, and known genealogy is given in Appendix A.

### 4.2. Phenotyping

The seed material was collected from field experiments conducted in Sofia, Bulgaria (42°41′ N, 23°19′ E) during three crop seasons (2013/14, 2016/17, and 2020/21), denoted by the year of harvest (2014, 2017, and 2021). For each growing season, the accessions were sown in a random design in double 1 m-long rows and two replications. The soil type at the experimental field is leached vertisol, pH 6.1, with 3.1% humus, 1420 mg total N per kg soil, of which 18 mg inorganic N. Plants received 120 kg N/ha as ammonium nitrate in two split doses, 40 kg/ha two weeks after sowing and 80 kg/ha before stem extension growth stage. For pest control, standard agricultural practices were applied. The average monthly temperature ranged between −5.8 °C and 23.2 °C, during the vegetation cycles (October–July). The monthly precipitation ranged from 4.7 mm to 150.5 mm, with average annual precipitation of 682 mm, which is around 20% higher than the climate norm for Sofia. Information on the monthly weather statistics for the region of the experimental field is presented in Appendix A.

Plant material was hand-harvested and hand-threshed. For each genotype and crop season, the thousand kernel weight (TKW) was determined based on grain number and grain mass of 10 main spikes per replication. The grain protein content (GPC) was measured in three pooled whole-grain powder samples (1.0 g each) for each accession and year. Before analyses, seeds were dried to a constant weight and ground to a fine powder with IKA Tube Mill Control (IKA Werke GmbH & Co., Staufen, Germany). The total N concentration was determined using a UDK 159 Automatic Kjeldahl Nitrogen Protein Analyzer (Velp Scientifica, Usmate, MB, Italy), and the corresponding GPC values (in %) were automatically inferred from the N values, according to [98].

### 4.3. Statistical Analyses

The normal distribution of the empirical data was checked by the Shapiro–Wilk test using the online software MVApp (https://mvapp.kaust.edu.sa) (accessed on 12 January 2024). The significant differences in GPC and TKW among accessions, growing seasons, and interaction effects between genotype and environment were tested using analysis of variance (ANOVA). The best linear unbiased estimators (BLUEs) for each accession across the growing seasons were obtained to eliminate the environmental impact by assuming the genotype as a fixed effect and the growing season as a random effect. The relationships for GPC and TKW among the growing seasons and with the BLUEs of these traits were tested by Pearson’s correlation coefficients (*r*). The correlation coefficients *r* were calculated from the empirical phenotypic data obtained in each growing season and from the mean BLUE values.

Broad sense heritability *h*^2^ for GPC and TKW was calculated using the formula
$$h^{2}=\frac{\sigma_{G}^{2}}{\left(\sigma_{G}^{2}+\frac{\sigma_{E}^{2}}{nE}\right)}$$
where $\sigma_{G}^{2}$ is the genotype variance, $\sigma_{E}^{2}$ is the variance of the residual, and *nE* is the number of environments (growing seasons). Heritability in each environment was estimated using the same formula, where *nE* in the denominator is the number of replications in a given environment.

All phenotypic data analyses were accomplished using STATISTICA 14 [99].

### 4.4. Association Mapping and Candidate Gene Search

Before performing marker-trait association analysis, the population stratification, the genetic relatedness among population entries, and the LD were considered. The genotypic data for the association panel of 179 accessions were already available from the 25K Infinium iSelect array (SGS Institut Fresenius GmbH TraitGenetics Section, Gatersleben, Germany) and described in [32]. Population structure was modeled using the Bayesian clustering algorithm in STRUCTURE 2.3.4, and the membership coefficients (Q-values) were determined [32]. Kinship was estimated as a similarity matrix (K) from 17,083 SNPs based on the method developed by VanRaden [100] using the Genomic Association and Prediction Integrated Tool (GAPIT) in R package. The values of LD decay (in Mbp) were determined for each chromosome as described in [32].

Considering the distinctive population structure and the high genetic relatedness among population individuals, GWAS was undertaken with a mixed linear model (MLM) Q + K model to control pseudo associations [101]. Hence, the filtered set of 17,083 SNPs, phenotypic data from the three crop seasons, and the calculated BLUE mean values for the traits, along with population structure (Q values) and kinship similarity matrix (K) as covariates were used for association mapping analysis in TASSEL v.5.

To analyze our data, we applied another powerful algorithm—the fixed and random model circulating probability unification (FarmCPU)—through GAPIT in R. The FarmCPU model uses the advantages of the MLM and stepwise regression (fixed effect model) and, at the same time, overcomes their disadvantages, which makes this approach very accurate for GWAS by effectively controlling the rate of false positive signals [102]. The FarmCPU algorithm delivered altogether three significant associations for the two traits, while the MLM output involved a large number of genomic signals. To test how the two models fit the data, we examined the quantile–quantile (Q-Q) plots, which show the distributions of important *p*-values (expected vs. observed −log_10_ (*p*-values)). Based on the number of significant SNPs (QTNs) and the Q-Q plots, we present here the results obtained by the MLM (Q + K) model. In order to reduce the number of false positive signals, we set the threshold of statistically significant QTNs at −log_10_ (*p*-value) > FDR (false discovery rate), at *p* < 0.01. FDR was calculated for each trait and environment.

To draw the Manhattan plots and Q-Q plots, R package qqman was employed [103].

Additive effects and R^2^ (percent phenotypic variation) of QTNs were estimated in TASSEL v.5 and Excel 16 (Microsoft). The percentage of phenotypic variation explained by each QTN (R^2^) was calculated as the difference of R^2^ with and without the strongest associated SNP.

For the significant trait-associated SNPs, flanking regions within the LD-estimated interval on either side of the marker were searched for candidate genes. Within these regions, high-confidence putative candidate genes were predicted by blasting against the cv. Chinese Spring reference genome IWGSC RefSeq v1.0. [104]. Gene annotations, gene ontologies (GOs), InterPros, and details for the potential candidate genes were obtained using EnsemblPlants, and Persephone web-based platforms (http://plants.ensembl.org/Triticum_aestivum/Info/Index; https://web.persephonesoft.com/?data) (accessed on 12 March 2024). 

## 5. Conclusions

The GWAS revealed significant QTNs underlying both studied traits. Among these, seven GPC-associated loci as well as four TKW-related loci had notably positive effect on the traits, thereby representing promising genomic regions for use in breeding projects aiming at improving grain protein and yield, respectively. While some of the loci confirmed already published ones, others might be new. In addition, to our knowledge, the targeted GPC-linked regions on chromosomes 2A, 7A, and 7B, as well as two TKW-associated intervals on chromosomes 1B and 6A, have not been previously linked to the studied traits. Based on their gene functional annotation, a few of the many high-confidence genes that were retrieved from the trait-associated genomic regions were selected as intriguing candidates. Annotations to senescence- and germination-related proteolysis, peptide chain elongation and protein translocation, synthesis of storage proteins, structural carbohydrates, and auxin, as well as a number of transcriptional regulatory factors and signal transducers, are present in these genes.

## Figures and Tables

**Figure 1 plants-13-01084-f001:**
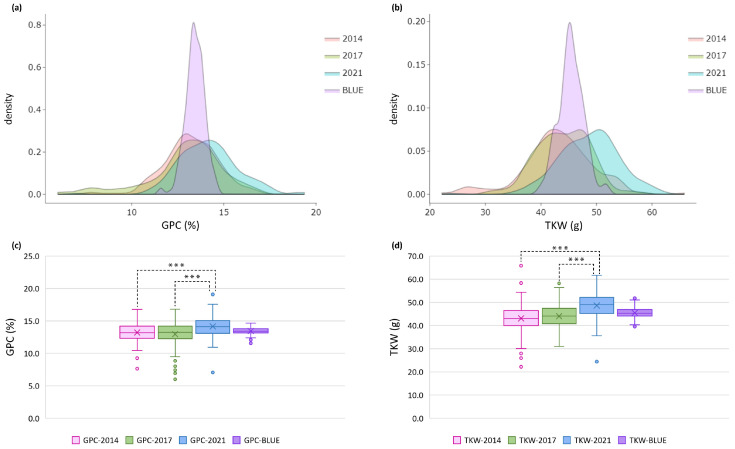
Probability density and box plots for (**a**,**c**) grain protein content (GPC) and (**b**,**d**) thousand kernel weight (TKW) across three growing seasons and based on best linear unbiased estimators (BLUEs) in a set of 179 Bulgarian bread wheat accessions. *** is the significant differences between means at *p* < 0.001.

**Figure 2 plants-13-01084-f002:**
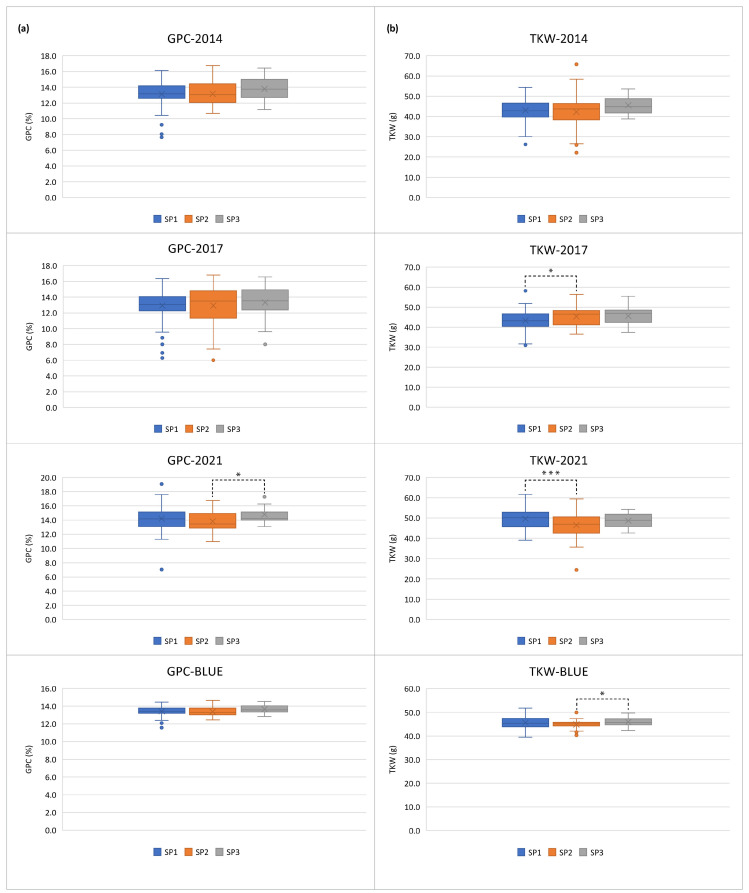
Box plots for (**a**) grain protein content (GPC, in %) and (**b**) thousand kernel weight (TKW, in g) across three growing seasons and based on the best linear unbiased estimators (BLUEs) in the three subpopulations (SPs) distinguished by the Q-matrix [32] within a population of 179 Bulgarian bread wheat accessions. *, *** are the significant differences between means at *p* < 0.05 and 0.001, respectively.

**Figure 3 plants-13-01084-f003:**
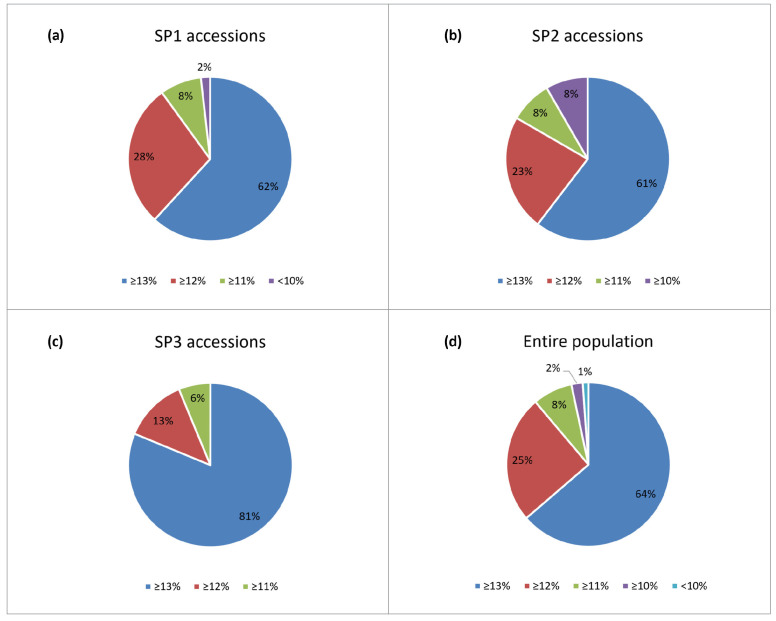
Proportion of accessions (**a**–**c**) from the three subpopulations distinguished by the Q-matrix [32] and (**d**) within the entire population of 179 Bulgarian bread wheat accessions with respect to grain protein content (GPC); classification is according to [2].

**Figure 4 plants-13-01084-f004:**
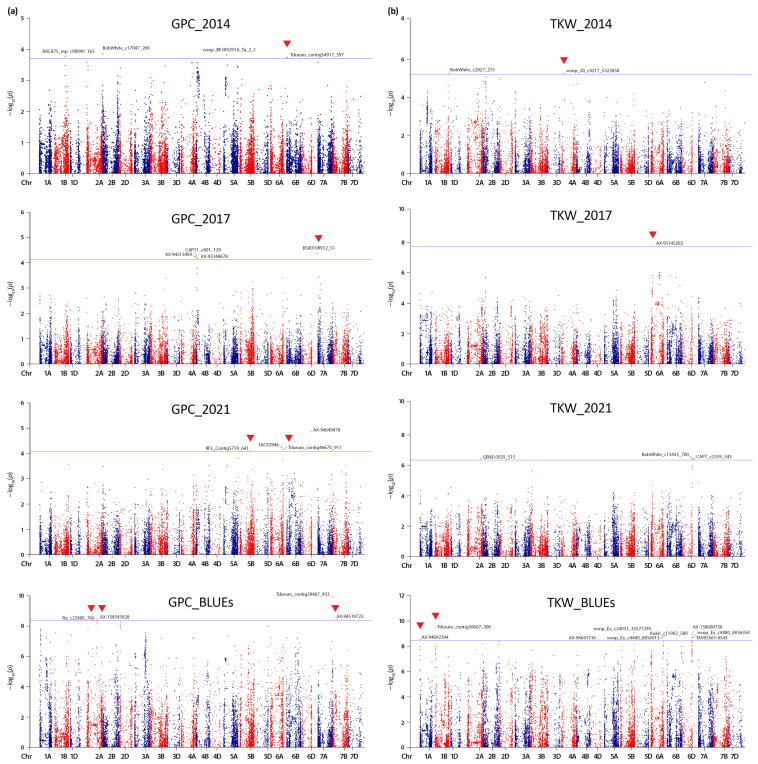
Manhattan plots of the genome-wide association scan for (**a**) grain protein content (GPC) and (**b**) thousand kernel weight (TKW) based on the MLM (Q + K) model and the phenotypic data from three growing seasons, along with the calculated BLUE values in a set of 179 Bulgarian bread wheat accessions. The blue color line depicts the threshold of –log_10_ (*p*) = FDR, determined for each trait and environment. The significant quantitative trait nucleotides (QTNs) are above the blue line. QTNs with positive additive effects on the traits are marked with red color triangles.

**Table 1 plants-13-01084-t001:** Descriptive statistics for grain protein content (GPC) and thousand kernel weight (TKW) in a set of 179 Bulgarian bread wheat accessions evaluated over three growing seasons.

Trait	Env.	Mean *	Std. Dev.	Min.	Max.	CV %	*h* ^2^
GPC (%)	Sofia 2014	13.2 b	1.46	7.6	16.8	11.09	0.64
	Sofia 2017	12.9 b	2.06	6.0	16.8	15.90	0.78
	Sofia 2021	14.2 a	1.62	7.1	19.4	11.43	0.69
	Average	13.4	1.34	9.4	16.8	10.00	0.82
	BLUE	13.4	0.51	11.6	14.7	3.81	
TKW (g)	Sofia 2014	43.1 b	6.43	22.2	65.8	14.92	0.77
	Sofia 2017	44.0 b	4.72	31.0	58.2	10.73	0.64
	Sofia 2021	48.6 a	5.42	24.4	61.6	11.14	0.70
	Average	45.2	4.28	33.7	55.0	9.46	0.81
	BLUE	45.4	2.25	39.6	51.8	4.94	

* different letters denote significant difference between the mean values at *p* < 0.001. Env. = environments; Std. Dev. = standard deviation; CV = coefficient of variation; *h*^2^ = broad-sense heritability; BLUE = best linear unbiased estimator.

**Table 2 plants-13-01084-t002:** Factorial analysis of variance (ANOVA) for (**a**) grain protein content (GPC) and (**b**) thousand kernel weight (TKW) across three environments (growing seasons) for a set of 179 Bulgarian bread wheat accessions.

(**a**)						
**Source of Variation**	**SS**	**df**	**MS**	**F**	***p*-Value**	**F Crit**
Genotype (G)	962.685	178	5.408	3.010	0.0000	1.233
Environment (E)	145.483	2	72.741	40.485	0.0000	3.021
G × E	639.638	356	1.797	7.210	0.0000	3.320
Total	1747.805	536				
(**b**)						
**Source of Variation**	**SS**	**df**	**MS**	**F**	***p*-Value**	**F Crit**
Genotype (G)	9792.257	178	55.013	2.897	0.0000	1.233
Environment (E)	3128.650	2	1564.325	82.375	0.0000	3.021
G × E	6760.582	356	18.990	4.3643	0.0000	4.092
Total	19681.49	536				

SS = sum-of-squares; df = degrees of freedom; MS = mean squares; F = F-value; F crit = F-critical value.

**Table 3 plants-13-01084-t003:** Pearson’s correlation coefficients (*r*) for grain protein content (GPC) and thousand kernel weight (TKW) among growing seasons (harvests in 2014, 2017, and 2021) and with the mean best linear unbiased estimator (BLUE) values in a set of 179 Bulgarian bread wheat accessions.

	GPC-2017	GPC-2021	GPC-BLUE	TKW-2014	TKW-2017	TKW-2021	TKW-BLUE
GPC-2014	0.69 ***	0.50 ***	0.93 ***	0.00	−0.09	−0.08	−0.05
GPC-2017		0.10	0.61 ***	0.04	0.01	−0.01	−0.02
GPC-2021			0.47 ***	0.16 *	−0.01	0.12	0.06
GPC-BLUE				−0.01	−0.10	−0.07	0.02
TKW-2014					0.39 ***	0.30 ***	0.60 ***
TKW-2017						0.53 ***	0.42 ***
TKW-2021							0.38 ***

*, *** significant at *p* < 0.05 and 0.001, respectively.

**Table 4 plants-13-01084-t004:** Quantitative trait nucleotides (QTNs) and candidate genes associated with grain protein content (GPC) detected in a population of 179 Bulgarian bread wheat accessions using the MLM (Q + K) model.

Marker	Chr	Position (bp)	−log_10_ (*p*)	Effect	R^2^	Gene ID	Annotation	Co-Located Loci ^a^
GPC_2014								
*RAC875_rep_c98040_163*	1B	512,632,139	3.78	−1.054	0.084	*TraesCS1B01G294600*	GTP cyclohydrolase II/3,4-dihydroxy-2-butanone 4-phosphate synthase	
*BobWhite_c17047_268*	2B	11,077,394	3.85	−1.011	0.089	*TraesCS2B01G023800*	Ethylene receptor	
*wsnp_BE500291A_Ta_2_1*	5A	148,056,504	3.82	−0.889	0.087	*TraesCS5A01G101200*	1-acyl-sn-glycerol-3-phosphate acyltransferase	
*Tdurum_contig54917_597*	6B	32,324,229	3.73	1.357	0.083	*TraesCS6B01G052900*	Kelch repeat–containing protein	[33]
GPC_2017								
*CAP11_c601_120*	4B	38,791,353	4.21	−2.456	0.096	*TraesCS4B01G050200*	Histidine kinase	
*AX-95148678*	4B	39,015,957	4.15	−2.446	0.095	*TraesCS4B01G050400*	UDP-glucose 6-dehydrogenase	
*AX-94513493*	4B	39,045,208	4.21	−2.456	0.096	*TraesCS4B01G050500*	UV-B-induced protein, chloroplastic	
*BS00109912_51*	7A	653,925	4.36	2.874	0.101	NA		
GPC_2021								
*RFL_Contig5739_641*	5B	531,538,634	4.11	1.412	0.092	*TraesCS5B01G350900*	Plant regulator RWP-RK family protein, putative	[34]
*IACX2946*	6A	599,046,570	4.16	−2.172	0.092	*TraesCS6A01G378000*	Protein kinase family protein	[23,33]
*Tdurum_contig46670_911*	6A	599,050,921	4.16	2.172	0.092	*TraesCS6A01G378100*	Diphthine-ammonia ligase	[23,33]
*AX-94649878*	6D	453,347,731	4.86	−1.954	0.111	*TraesCS6D01G362900*	NAC domain protein	
GPC_BLUE								
*Ra_c22880_760*	2A	498,105,752	8.38	7.918	0.217	*TraesCS2A01G289800*	Alpha-amylase	
*AX-158545828*	2A	501,850,514	8.46	8.069	0.219	*TraesCS2A01G291200*	Mitochondrial import inner membrane translocase subunit TIM22	
*Tdurum_contig59467_433*	7A	715,692,069	9.94	−9.349	0.267	NA		
*AX-94519723*	7B	121,890,360	8.57	7.372	0.222	*TraesCS7B01G105700*	DNA-binding storekeeper protein-related transcriptional regulator	

^a^ Previously reported significant marker/QTL in close proximity to the QTNs identified in the current study.

**Table 5 plants-13-01084-t005:** Quantitative trait nucleotides (QTNs) and candidate genes associated with thousand kernel weight (TKW) detected in a population of 179 Bulgarian bread wheat accessions using the MLM (Q + K) model.

Marker	Chr	Position (bp)	−log_10_ (*p*)	Effect	R^2^	Gene ID	Annotation	Co-Located Loci ^a^
TKW_2014								
*BobWhite_c2027_215*	1B	640,555,504	5.27	−10.268	0.124	*TraesCS1B01G415400*	TSA: Wollemia nobilis Ref_Wollemi_Transcript_14910_3291 transcribed RNA sequence	[21,35]
*wsnp_JD_c4217_5322858*	4A	140,291,399	5.25	8.388	0.123	*TraesCS4A01G115300*	Small nuclear ribonucleoprotein	[36]
TKW_2017								
*AX-95145282*	6A	100,766,008	7.84	4.780	0.199	NA		
TKW_2021								
*GENE-0925_515*	2A	656,820,630	6.42	−4.526	0.143	NA		
*BobWhite_c13435_700*	6D	471,008,534	6.43	−4.534	0.144	NA		[37]
*CAP7_c2559_543*	6D	471,017,962	6.40	−4.519	0.143	*TraesCS6D01G402800*	Zinc finger CCCH domain-containing protein 19	[37]
TKW_BLUEs								
*AX-94692394*	1A	7,186,246	8.54	1.927	0.209	*TraesCS1A01G012600*	Leucine-rich repeat receptor-like protein kinase family protein	[21,35]
*Tdurum_contig50667_306*	1B	20,588,032	9.49	2.705	0.238	*TraesCS1B01G041200*	F-box protein-like	
*AX-94641316*	4D	374,832,658	8.55	−2.485	0.210	*TraesCS4D01G219400*	Regulator of Vps4 activity in the MVB pathway protein, putative	
*wsnp_Ex_c24031_33277293*	5B	707,131,836	9.10	−3.002	0.226	*TraesCS5B01G560400*	PGR5-like protein 1A, chloroplastic	[35,37]
*wsnp_Ex_c4480_8056013*	6A	611,487,998	8.76	−2.084	0.216	*TraesCS6A01G405500*	Lysine-specific histone demethylase 1-3-like protein	[35,37,38]
*Kukri_c11902_580*	6A	611,491,408	8.76	−2.085	0.216	*TraesCS6A01G405500*	Lysine-specific histone demethylase 1-3-like protein	[35,37,38]
*wsnp_Ex_c4480_8056354*	6D	464,946,552	8.67	−2.078	0.214	*TraesCS6D01G389200*	Lysine-specific histone demethylase 1-3-like protein	[37]
*AX-158600736*	6D	464,948,739	8.70	−2.078	0.215	*TraesCS6D01G389200*	Lysine-specific histone demethylase 1-3-like protein	[37]
*TA005561-0543*	6D	465,050,368	8.48	−2.065	0.211	NA		[37]

^a^ Previously reported significant markers/QTL coinciding with (underlined references) or in close proximity to the QTN identified in the current study.

## Data Availability

Data are contained within the article and its Appendix A.

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
