# Peer review of "GWAS in a Collection of Bulgarian Old and Modern Bread Wheat Accessions Uncovers Novel Genomic Loci for Grain Protein Content and Thousand Kernel Weight"

_plants, 2024, doi:10.3390/plants13081084_

Round 1

Reviewer 1 Report

Comments and Suggestions for Authors

1. Please provide high-resolution figures for Figure 1 and add statistical analysis for Fig. 1C and D.

2. What is the mean for SS, df, MS, F, and F crit in Table 2? The authors should add detailed information for this abbreviation.

3. Please provide high-resolution figures for Figure 2 and more detailed information for SP1, SP2 and SP3. And delete (B) in the figure legend. 

4. The resolution for Figs. 3-5 is low; please replace high-resolution ones. 

Reviewer 2 Report

Comments and Suggestions for Authors

The authors present a study into the genetic control of grain protein content and thousand kernel weight in Bulgarian wheat germplasm. The manuscript is generally well-written but appears to suffer from a high-rate of false positives which reduces its utility.  

The relatively small sample size (for an association mapping study in wheat) and large linkage blocks cause some issues with the study, particularly the inflation of the false positive discovery rate. This resulted in a large number of significant associations. 

The Q-Q plots show poor fit despite using a MLM+Q+K model, suggesting that population structure was not adequately controlled. 

The high percentage of variation explained by various QTL makes me suspect this is either a factor of the small population size or a result of population stratification inflating R2 values.

There are a number of options the authors can explore to improve their model fits and false discovery rate.

1) Consider pruning markers based on LD prior to running association analyses.

This is important for the population structure analysis, as highly correlated markers can inflate STRUCTURE results. If the structure analysis is biased the model fit will be poor.

Thinning markers based on haploblocks of highly correlated markers may reduce spurious associations. Consider R2 filtering in PLINK or equivalent software packages. This should help reduce the number of putatively false associations. 

2) Consider alternative models, have the authors considered MLM+PCA+K or FarmCPU models to control for false positives?

3) Consider blocking factors in their mixed model as covariates (i.e. block based on origin, contemporary versus historic accessions) to improve model fit.

4) Employ a false discovery rate control that accounts for marker correlation. Bonferroni's method does not take into account marker linkages. The FDR adjust p-value (q-value) method may be more appropriate.  

Reviewer 3 Report

Comments and Suggestions for Authors

Review of the manuscript:

Manuscript "GWAS in a collection of Bulgarian old and modern bread wheat accessions uncovers novel genomic loci for grain protein content and thousand kernel weight"

The aim of the study was to identify promising genomic regions for wheat improvement and to delineate candidate genes. For this purpose, the authors used GWAS. This is not an original idea, although the method used can give interesting results.

May fill the gap in the knowledge of bread wheat varieties. GWAS method analyzes genotypes with known similarity/relatedness. M reviewed manuscript, unfortunately, no information about the relatedness of genotypes was provided. This should be completed.

Besides, the authors compare old and new varieties. However, a comparative analysis of the two groups based on marker observations is not provided. This should be supplemented by providing a similarity/differentiation structure.

Why are results from selected years (2014, 2017 and 2021) given and not all from 2014-2021?

Chapter 4.2 describes phenotyping. However, the methodology for DNA isolation and genotyping is not presented. It is imperative that this be supplemented.

The Authors demonstrated the significance of GE interactions. The manuscript should be supplemented with weather conditions and a comparative analysis of the results obtained in the context of varying weather conditions.

The quality of the Figures is very poor. They should be improved.

In the notation of the decimal logarithm, the number "10" is not written. It is imperative that this be corrected throughout the manuscript. On the Figures as well.

To be completed:

Testing the consistency of the empirical distribution with the normal distribution.

Provide information on which data the correlation coefficients were calculated from.

To supplement statistical analyses with multivariate similarity: analysis of canonical variables, Mahalanobis distances.

Paper needs major revision.

Round 2

Reviewer 3 Report

Comments and Suggestions for Authors

The Authors revised the manuscript significantly according to my comments and suggestions. The only remaining issue is the correctness of the notation of the logarithm with base 10.
Figure 4: The logarithm with base 10 notation does not specify the number 10. It should be: "-log(p)" - without '10'. This should be corrected on the figures.
Table 4: Similar to Figure 4.
Table 5: Similar to Figure 4.

Paper needs minor revision.
